# Tribological Behavior of High Entropy Alloy Coatings: A Review

Dawei Luo [1], Qing Zhou [1,2,*], Zhuobin Huang [1], Yulong Li [2,*], Yulin Liu [1], Qikang Li [1], Yixuan He [1] and Haifeng Wang [1,*]

1   State Key Laboratory of Solidification Processing, Center of Advanced Lubrication and Seal Materials, Northwestern Polytechnical University, Xi'an 710072, China
2   Institute for Applied Materials (IAM), Karlsruhe Institute of Technology (KIT), Kaiserstrasse 12, 76131 Karlsruhe, Germany
*   Correspondence: zhouqing@nwpu.edu.cn (Q.Z.); yulong.li@partner.kit.edu (Y.L.); haifengw81@nwpu.edu.cn (H.W.)

**Abstract:** As engineering equipment is applied in a harsh environment with a heavy load, cyclic stress, and a wide range of temperatures, the reliability of the equipment becomes a challenge, especially when wear contact is involved. Hence, the design and exploitation of an advanced alloy surface may hold the key to control and minimize friction and wear in the transmission system for safety-critical applications. High entropy alloys (HEAs) or multi-component alloys have been proved to have outstanding mechanical properties, corrosion resistance, and high-temperature oxidation resistance with potential use as wear resistance and friction reduction coatings. In this paper, the properties and development status of HEAs coating systems for tribological applications were reviewed to gain a better understanding of their advantages and limitations obtained by different preparation methods. Specifically, focus was paid to magnetron sputtering, laser cladding, and thermal spraying since these three deposition methods were more widely used in wear-resistant and friction-reducing coatings. Building upon this, the correlation between composition, mechanical properties, and friction as well as wear characteristics of these coatings are summarized. Finally, the key problems to be solved to move the field forward and the future trend of tribology application for HEA coatings are outlined.

**Keywords:** high entropy alloy; coatings; mechanical property; tribology

## 1. Introduction

Friction and wear inevitably occur in mechanical systems in operation, which can lead to the degradation of key transmission components and thus the failure of mechanical equipment. Lots of research has been done to develop anti-friction and wear-resistant materials for working in different environments [1,2]. With the further exploration of the universe, polar region, and deep sea, the service conditions of high load, broad temperature change, and intense irradiation of the equipment put forward higher requirements for material performance.

Since Yeh and Cantor et al.'s pioneering work of defining alloy comprising at least five principal elements with each composition range between 5% and 35% as high entropy alloys (HEAs), it has attracted extensive attention due to their appealing properties and potential applications [3,4]. For such an alloy, due to their high configurational entropy, sluggish diffusion, and serious lattice distortion, it is often considered that a simple solid-solution phase with face-centered cubic (FCC), body-centered cubic (BCC), or hexagonal close-packing (HCP) structures, instead of general intermetallic compounds, would be observed. The unique structure endows HEAs with better properties than traditional alloys, such as high strength and hardness [5], high-temperature oxidation resistance [6], high corrosion resistance [7], fatigue resistance, and fracture toughness [8]. The coupling of many outstanding properties endows HEAs with potential value as wear-resisting and friction-reducing materials [9,10]. In our previous work, the influence of Al content on the

microstructure and wear resistance of $Al_xTiZrNbHf$ refractory high entropy alloys (RHEAs) was studied. With the increase of Al content, the $Al_xTiZrNbHf$ RHEAs appeared another BCC phase, the hardness increased from 229 to 420 HV, and the wear volume decreased from 0.065 to 0.045 $mm^3$ [11]. Pei et al. reported that the extraordinary wear resistance of $TiZrV_{0.5}Nb_{0.5}Al_{0.5}$ RHEAs at temperatures from 600 to 800 °C, with wear rates as low as $10^{-6}$ $mm^3$ $N^{-1}$ $m^{-1}$, which is one of the most wear-resistant alloys reported at this temperature [12]. However, the cost of preparing HEAs is much higher than that of the traditional alloy due to its inclusion of expensive elements (such as Hf, Nb, Zr, V, Ti, Mo, W, etc.). Therefore, HEA coatings and films with large coverage area and less material consumption have gradually entered the sight of material community.

In the past decade, various techniques have been employed for depositing HEA coatings and films, e.g., laser cladding [13,14], magnetron sputtering [15,16], thermal spraying [17,18], electrochemical deposition [19,20], and others. According to the existing results, HEA coatings not only have higher hardness and strength than bulk HEAs, but also have excellent performance in radiation resistance [21,22], high-temperature oxidation resistance [23–25], corrosion resistance [26,27], and superior wear resistance [28,29]. In this paper, advancements made in the fabrication and development of HEA coatings, and films for friction and wear reduction were reviewed. The objective is to provide a comprehensive overview of HEA coatings/films in tribological applications and to better understand their advantages and limitations.

## 2. Preparation Technologies

In this section, the methods used to prepare HEA coatings are summarized, and the advantages/disadvantages of each method are discussed. Up to now, laser cladding, magnetron sputtering, thermal spraying, electrochemical deposition, and other methods have been widely used in the deposition of HEA coatings. Due to the high cooling rate of the preparation method, the microstructure of the HEA coatings is obviously different from that of the bulk HEAs, which is easy to form amorphous and nanocrystalline structures, thus achieving higher hardness and strength [30]. Of course, the insufficient diffusion caused by high-speed cooling will inhibit the nucleation and growth of intermetallic compounds, which is more conducive to forming simple solid-solution phases.

### 2.1. Laser Cladding

Laser cladding has a high heating and cooling rates, so it has a larger degree of undercooling in the synthesis process. This kind of process would inhibit element diffusion and avoid component segregation, resulting in a more uniform structure. The solid binding strength provided by metallurgical bonding of the cladding layer and melting zone is much higher than magnetron sputtering and thermal spraying technique. Zhang et al. fabricated a FeCoNiCrCu HEA coating by laser cladding. Obviously, the component of this coating is more uniform than the conventional casting multi-element alloys [31]. Under a large temperature gradient, the coating changed from columnar to equiaxed grains, as shown in Figure 1.

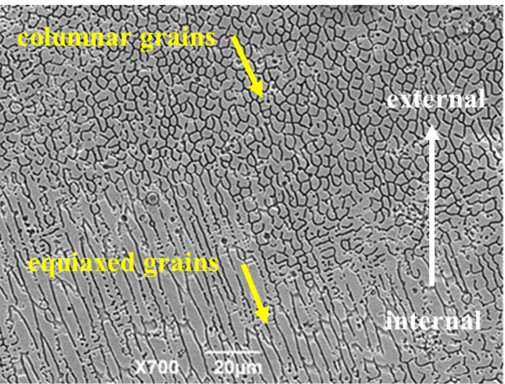

**Figure 1.** Microstructure of the cross-sectional FeCoNiCrCu coating prepared by laser cladding [31].

The melting of the base material during laser cladding will induce its elements into the coating, hence causing the change in the composition of this layer, which can be called the dilution rate. Dilution is inevitable, but on the other hand, in order to maintain the bonding strength between the sediment and matrix, it is necessary to improve metallurgical bonding by adjusting the dilution rate. Generally speaking, the appropriate dilution rate should be below 15% [32]. The HEA coating prepared by laser cladding will be significantly affected by a large number of components and significant differences in melting points of each component, so the control of dilution rate is particularly challenging and critical. Cai et al. found that when FeCrCoNi powder was deposited on the surface of Cr12MoV steel, it is much easier to change dilution rate via designing the composition of feedstock powder than tuning the process parameters [33]. Specifically, it is demonstrated that the dilution rate is proportional to the Fe content. Yue et al. deposited an AlCoCrCuFeNi coating on pure magnesium plate by laser cladding and found that the Mg-Cu phase could be obtained by the diffusion of Cu into the Mg matrix [34]. Except for this change in the Cu content of the HEA, no remarkable dilution of the HEA composition occurred.

### 2.2. Magnetron Sputtering

Magnetron sputtering adds a closed magnetic field to the sputtering target, effectively prolonging the motion of electrons, reducing the scattered atoms, and improving the ionization efficiency. It can be divided into DC magnetron sputtering, radio frequency magnetron sputtering, reactive magnetron sputtering, and bias magnetron sputtering. Among them, DC sputtering is suitable for materials with good electrical conductivity; radio frequency sputtering is mainly used for insulation or poor electrical conductivity of non-metallic materials; reactive sputtering is used for the preparation of alloy oxides, carbides, and nitride; and bias sputtering can improve the density of the film, as well as increase the bonding force of the film.

HEA films prepared by magnetron sputtering can be divided into two categories according to the form of targets. Frist, HEA films were deposited directly by an HEA as a target, which can guarantee the consistency of the chemical composition of HEA films. However, the preparation for HEA target is time consuming and its composition is not flexible to be tuned. Wang et al. prepared the CoCrFeMnNi alloy target by vacuum arc melting [35]. Using radio frequency magnetron sputtering, different thicknesses of CoCrFeMnNi HEA films were deposited on Si (100) substrates. It is concluded that the thin film has excellent fatigue resistance (~$10^6$ cycles) and high hardness (~8.5 GPa), which is caused by the synergistic effect of strain hardening and detwinning of ultra-high-density nano twins.

Alternatively, HEA films can be fabricated by co-deposition of polymetallic targets. This technology can avoid the complex preparation process of alloy target, but it is difficult to obtain the HEA films with ideal structure by changing the deposition parameters, and the amorphous structure is easily formed. Changing deposition parameters to obtain the desired element composition of the film is an even greater challenge. Zou et al. prepared NbMoWTa HEA film by co-deposition method and then cut the film into nano-pillars by focusing the ion beam, as shown in Figure 2 [36]. The size effect of NbMoWTa in terms of grain size and the cylinder diameter is obtained by compression test. After annealing at 1100 °C for 72 h, the grain size of the film still remained at 70 to 100 nm, showing good thermal stability.

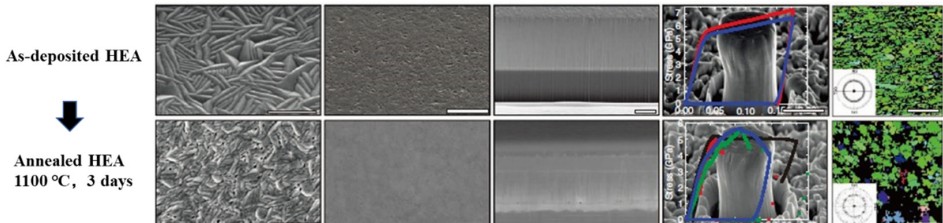

**Figure 2.** Comparison of the microstructures of NbMoWTa refractory high entropy alloy films after 72 h heat treatment at 1100 °C (scale bars, the first column 200 nm, last column 300 nm, other images 1 µm) [36].

## 2.3. Thermal Spraying

Thermal spraying is a kind of technology of forming lamellar coating by heating and spraying material to a molten or semi-molten state by the heat source and mechanically combining it with the substrate surface by high-pressure gas. At present, the thermal spray coating methods used to prepare the HEA coating mainly include plasma spraying coating (PS), high-velocity oxygen-fuel spraying (HVOF), and cold spraying (CS). As early as 2004, when the HEA alloy was just defined, Huang et al. prepared two kinds of coatings (AlSiTiCrFeCoNiMo$_{0.5}$ and AlSiTiCrFeNiMo$_{0.5}$) by PS and found that the prepared HEA coating had both excellent oxidation and wear resistance [37]. It laid a foundation for the subsequent research of HEA coatings prepared by thermal spraying. Srivastava et al. fabricated a FeCoCrNi$_2$Al HEA coating by HVOF and investigated the microstructure, surface morphology, and properties of the coating [38]. The results show that the coating not only has a good adhesion with surface microhardness of 610 ± 30 VHN, and bearing capacity of 9.8 N, but also the coating has good high-temperature corrosion resistance as well, so the coating can be used as thermal barrier material or high-temperature anti-corrosion material. Yin et al. prepared a FeCoNiCrMn HEA coating by cold spraying and studied the microstructure and friction properties of the coating [39]. The experimental results show that the HEA coating prepared by CS has very low porosity and retains the HEA phase structure completely without phase transformation. As shown in Figure 3, the grain structure of HEA coating is refined due to the increase in dislocation density and dynamic recrystallization. Finally, compared with the coating prepared by laser cladding, the thermal spraying coating shows better wear resistance.

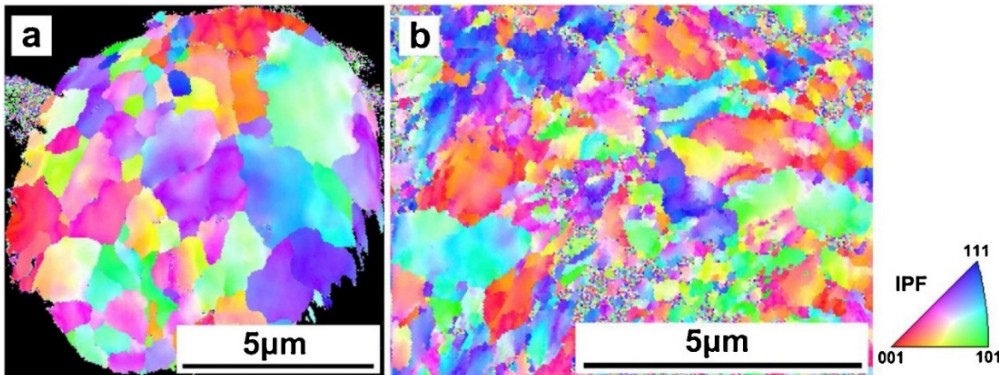

**Figure 3.** EBSD IPF maps of (**a**) a single HEA particle and (**b**) the HEA coating prepared by CS [39].

## 2.4. Electrochemical Deposition

Electrochemical deposition provides a possibility for the low-cost synthesis of HEA films, which does not require complex and expensive equipment and can also coat large areas of materials with complex surface morphology. Electrodeposition can easily control the element composition and morphology of the coating, so the study of electrodeposition of HEA has gain considerable attention due to its application prospect for preparing high entropy alloy coating.

Yoosefan et al. prepared a CoCrFeMnNi coating on copper substrate by pulse electro-chemical deposition and studied its wettability [40]. The results showed that all coatings were completely hydrophilic; the largest wettability angle (56°) is obtained for the as-prepared film at a frequency of 5000 Hz. Popescu et al. prepared CoCrFeMnNi high entropy alloy (HEA) thin films on Cu substrates by potentiostatic electrodeposition and studied their micro-structures and corrosion resistance [41]. The results show that the dense and uniform film is composed of 50 nm~5 μm particles and provides additional protection to the copper electrode during immersion in artificial seawater.

### 2.5. Other Techniques and Brief Summary of Preparation Technologies

Gao et al. deposited a CoCrFeNiMn coating on grey cast iron through plasma transfer arc claddings and investigated the relationship between its microstructure and corrosion resistance [42]. Pogrebnjak et al. deposited the TiHfZrVNb(N) coatings on steel substrate by the cathodic-arc-vapor deposition method and studied their tribological properties [43]. The results show that the enhanced hardness at higher than 40 GPa (TiHfZrVNb(N)) can effectively reduce the wear factor ($0.039 \times 10^{-5}$ mm$^3$ H$^{-1}$ mm$^{-1}$). Chandrakant et al. deposited an AlCoCrFeNi coating on AIAI410 stainless steel substrate by electro spark deposition and studied its mechanical properties and wear resistance [44]. The results show a threefold increase in the microhardness was achieved after coating, and the coating exhibited excellent wear resistance compared to steel.

The process characteristics of laser cladding, magnetron sputtering, thermal spraying, and electrochemical deposition are summarized, as shown in Table 1.

**Table 1.** Technology characteristics of laser cladding, magnetron sputtering, thermal spraying, and electrochemical deposition [13,15,17,45,46].

| Methods | Advantages | | Defects | |
|---|---|---|---|---|
| Laser cladding | (1)<br>(2)<br>(3) | Excellent bonding strength with substrate;<br>High heating and cooling rates;<br>High deposition rate; | (1)<br><br>(2)<br>(3) | Matrix elements dissolve into the film inevitably;<br>Difficult to modulate microstructure;<br>High coating roughness; |
| Magnetron sputtering | (1)<br><br>(2)<br><br>(3) | Flexible adjustment of process to control the composition and properties;<br>The thin film is with uniform composition and good density;<br>Nitride/oxide/carbide films can be achieved by inletting reactive gas; | (1)<br>(2)<br>(3) | Limited film thickness;<br>Low utilization of target material;<br>The roughness of the substrate surface is required; |
| Thermal spraying | (1)<br>(2)<br>(3)<br><br>(4) | High deposition efficiency;<br>The cost is low;<br>The spraying process has little effect on the base material;<br>Spraying materials are widely used; | (1)<br><br>(2) | Easy to form defects such as oxide, holes, cracks, and delamination;<br>Poor bonding strength with substrate; |
| Electrochemical deposition | (1)<br><br>(2)<br>(3) | The films can be evenly deposited on a substrate with a complex structure;<br>Low cost and low energy consumption;<br>Coating composition is easy to control. | (1)<br><br>(2)<br>(3) | It is difficult to prepare multilayer films with a complex structure;<br>Poor bonding strength with substrate;<br>Poor uniformity of film due to fluctuation of current and voltage. |

## 3. Composition and Microstructures

### 3.1. Prediction for Phase Structure

The phase structure of the HEA film is similar to that of the bulk alloy except for the amorphous and nanocrystalline structure due to rapid cooling, which can be predicted by a thermodynamic law [30]. Zhang et al. summarized the phases of high entropy alloys from the aspects of atomic radius difference ($\delta$), ($\Delta H_{mix}$), and entropy of mixing ($\Delta S_{mix}$) through statistical analysis of a large number of HEAs and proposed a new criterion based on the relationship between the parameters of mixing entropy and mixing enthalpy ($\Omega$): [47–49]

$$\Omega = T_m \Delta S_{mix} / |\Delta H_{mix}| \tag{1}$$

$$T_m = \sum_{i=1}^{n} c_i (T_m)_i \tag{2}$$

Here, $T_m$ is the average melting temperature of $n$-elements alloy and $(T_m)_i$ is the melting point of the $i$th component of the alloy. According to Boltzmann's hypothesis, the ideal configurational entropy of mixing of an $n$-element regular solution is as follows:

$$\Delta S_{mix} = -R \sum_{i=1}^{n} (c_i \ln c_i) \tag{3}$$

Here, $R$ (= 8.314 $JK^{-1}$ $mol^{-1}$) is the universal gas constant, and $c_i$ is mole percent of the $i$th component.

The enthalpy of mixing for a multi-component alloy system with $n$ elements can be determined from the following equation:

$$\Delta H_{mix} = \sum_{i=1, i \neq j}^{n} \Omega_{ij} c_i c_j \tag{4}$$

Here, $\Omega_{ij}$ (= $4\Delta H_{AB}^{mix}$) is the regular solution interaction parameter, and $\Delta H_{AB}^{mix}$ is the enthalpy of mixing of binary liquid alloys. Based on the available data of HEA, the empirical criterion for solid solution formation is $\Omega \geq 1.1$ and $\delta \leq 6.6\%$ [48]. Here, $\delta$ could be expressed as follows:

$$\delta = \sqrt{\sum_{i=1}^{n} c_i (1 - r_i/\bar{r})^2} \tag{5}$$

where $r_i$ is the atomic radius of the $i$th element, and $\bar{r}$ is the mean atomic radius (= $\sum_{i=1}^{n} c_i r_i$).

Of course, there are other criteria for the formation of HEAs. For example, Wang proposed a new atomic size difference criterion ($\gamma$) based on the Hume–Rothery rule. When $\gamma < 1.175$, the alloy is more likely to form a solid solution [50]:

$$\gamma = \left( \left( 1 - \sqrt{\left( (r_s + \bar{r})^2 - \bar{r} \right) / (r_s + \bar{r})^2} \right) / \left( 1 - \sqrt{\left( (r_l + \bar{r})^2 - \bar{r} \right) / (r_l + \bar{r})^2} \right) \right) \tag{6}$$

where $r_s$ and $r_l$ are the atomic radii of the smallest and largest atoms in the alloy, respectively.

After the solid solution structures are determined by parameters and $\delta$, the BCC and FCC structures need to be distinguished by VEC parameters [51,52].

$$VEC = \sum_{i=1}^{n} c_i (VEC)_i \tag{7}$$

where $(VEC)_i$ is the $VEC$ value for the $i$th element. Fcc phases are found to be stable at higher $VEC$ ($\geq 8$), and instead, BCC phases are stable at lower $VEC$ ($<6.87$).

### 3.2. HEA Coatings

Because of a thickness generally over 10 μm, the deposited HEA layers through laser cladding or thermal spraying techniques can be indicated as HEA coatings. HEA coatings are mainly classified into 3D transition metal-based HEA coatings and refractory metal HEA coatings [53]. The transition metal HEA coating is mainly focused on the CoCrFeNi alloy. By adding Al, Ti, Nb, Mo, and other alloying elements to adjust the phase and microstructure, the mechanical and functional properties of the coating can be improved. Ye et al. studied the effect of Al content on microstructure and wear resistance of the CoCrFeMnNi HEA cladding layer [54]. The results showed the dendritic microstructure of the $Al_x$CoCrFeMnNi cladding layer. When the content of Al was x $\leq$ 0.5, the cladding layer was still a single FCC phase; when x $\geq$ 1.0, the cladding layer was composed of two phases of FCC and BCC. The microhardness and wear resistance of the cladding layer increase with the increase of Al content in accordance with Archard's law. Xiang et al. prepared a CoCrFeNiNb$_x$ (x = 0, 1) HEA coating on a pure Ti matrix by pulse laser cladding [55]. Due to the size effect, the addition of Nb improves the solution strengthening effect and promotes the formation of the $Cr_2Ti$ Laves phase and $Cr_2Nb$ Laves phase with high hardness, so that the hardness of the alloy reaches 1008 HV. Xiao et al. prepared a series of FeCoNiCrSiAl$_x$ high entropy alloy coatings by plasma spraying and investigated their microstructure and wear resistance [56]. The microstructure of as-sprayed FeCoNiCrSiAl$_x$ HEA coatings is mainly the BCC phase with a minor amount of the FCC phase and is independent of Al content, as shown in Figure 4a. However, after heat-treated at 800 °C for 2 h, the main phases of FeCoNiCrSiAl$_x$ (x = 0.5, 1.0, 1.5) HEA coatings are FCC + $Cr_3Ni_5Si_2$, FCC + BCC + $Cr_3Ni_5Si_2$, and BCC, as shown in Figure 4b. A long heat treatment results in the formation

of intermetallic compounds in the coating, while the existence of the Al element inhibits this phenomenon, indicating that Al improves the phase stability of coating.

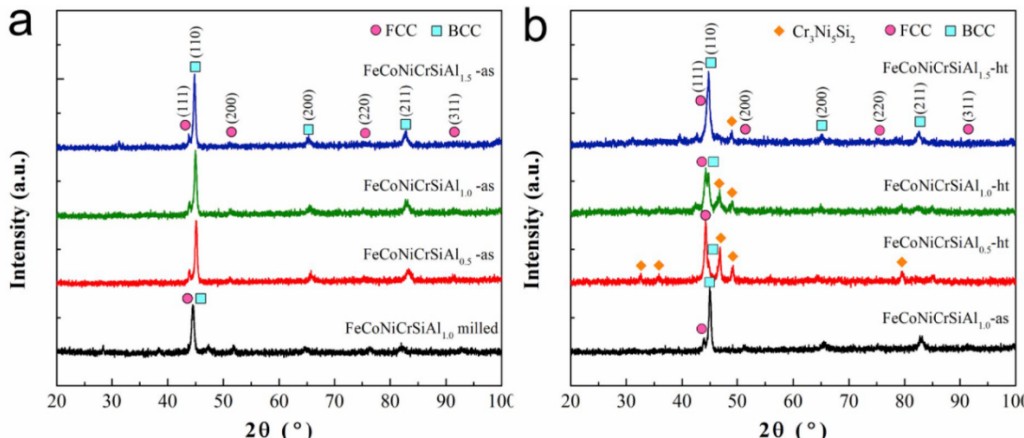

**Figure 4.** XRD patterns of the plasma-sprayed FeCoNiCrSiAl$_x$ HEA coatings: (**a**) as sprayed and (**b**) heat treated at 800 °C for 2 h [56].

Same to the bulk HEAs, the refractory coatings are composed of high melting point alloying elements, such as Nb, Zr, Ta, V, W, Hf, Mo, etc. Huang et al. prepared a novel refractory HEA coating with a composition close to TiNbZrMo on 316L steel surface by the alaser cladding method [57]. The results show that TiNbZrMo refractory HEA is mainly composed of a single BCC solid solution with a dendritic and interdendritic microstructure. Guo et al. deposited a series of MoFe$_x$CrTiWAlNb$_y$ refractory HEA coatings on M2 high-speed steel by laser cladding and studied the microstructure and mechanical properties of the coatings before and after annealing [58]. The coatings before annealing are mainly composed of BCC solid solution, MC carbide, and C14 Laves phase, and the phase structure remains unchanged when Fe and Nb contents change. After annealing at 650 °C, the dendrites and MC carbides were unaltered, but the C14 Laves phase gradually precipitated from the BCC matrix to improve the hardness of the coatings. In order to improve the hardness and high-temperature oxidation resistance of Ti-6Al-4V, Chen et al. prepared AlTiVMoNb refractory HEA coating by laser cladding on Ti-6Al-4V surface [59]. The results show that the AlTiVMoNb coating improves the hardness of Ti-6Al-4V surface by 2.5 times compared to 888HV, and the oxidation resistance is improved by nearly 10 times, which meets the good experimental expectations.

### 3.3. HEA Films

Compared with the thicker coatings, the thickness of HEA films is between several hundred nanometers to several microns, and the preparation methods generally adopted are magnetron sputtering and electrochemical deposition. Hu et al. studied the effect of substrate temperature on microstructure and mechanical properties of FeCoNiCrMn HEA thin films prepared by magnetron sputtering [60]. As shown in Figure 5, the 293 K-film has a cauliflower-like microstructure and is composed of small particles with an average particle size of about 30 nm, with nano cracks at the boundary. With the increase of substrate temperature, the nano-crack disappears, and the microstructure becomes fuzzy. As the temperature rose to 673 K, a unique morphology with facet feature was clearly seen, indicating an evident preferred-orientation grain growth of the FCC phase; grain coarsening occurs with further temperature increase to 773 K. It can be seen from Figure 5f that the hardness of the film increases first and then decreases with the increase of the substrate temperature. The increase in hardness is attributed to the disappearance of pores and the enhancement of interface bonding, and the latter decrease is due to the increase in grain size. In addition to the influence of preparation process on the film, there is also the influence of composition.

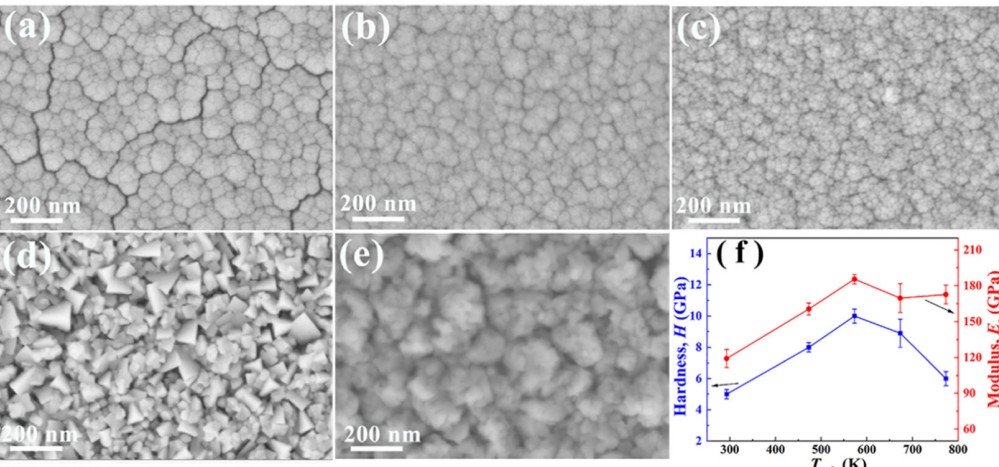

**Figure 5.** Surface morphology of HEA thin film FeCoNiCrMn at different substrate temperatures, (**a**) 293 K, (**b**) 473 K, (**c**) 573 K, (**d**) 673 K, and (**e**) 773 K. (**f**) The hardness and elastic modulus of FeCoNiCrMn films as a function of substrate temperatures [60].

Cheng et al. prepared $(TiZrHf)_x(NbTa)_{1-x}$ (x = 0.07~0.90) series of HEA films by co-deposition of TiZrHf and NbTa and investigated the relationship between the microstructure and mechanical properties [61]. All the as-deposited HEA thin films show a solid-solution BCC structure. As the compositional ratio (x) increases, the elastic modulus decreases from 153 to 123 GPa, following the trend of the rule of mixture. The thickness of the film also significantly affects its performance. Feng et al. prepared NbMoTaW HEA films with a thickness ranging from 100 to 2000 nm and studied their microstructure stability and mechanical properties [62]. The results showed that there was no obvious change in the phase after annealing at 800 °C for 2 h. The hardness increases monotonously with the decrease of film thickness, showing a strong size dependence. The design of interface structure also has a significant effect on the microstructure and properties of thin films. Wu et al. designs a laminate composite multilayer with alternating CoCrNi (HCP) and TiZrNbHfCrCoNi (amorphous) layers [63]. It is shown that the elemental partitioning among adjacent amorphous and crystalline phases of the multilayer leads to their mutual thermodynamic and mechanical stability and greatly improves the yield strength of the original TiZrNbHf-based amorphous phase.

Compared with HEA films deposited by magnetron sputtering, electrochemically deposited films are more likely to produce a large number of microcracks due to insufficient bonding force. Soare et al. deposited AlCrFeMnNi and AlCrCuFeMnNi HEA films in an electrolyte based on a DMF (N,N-dimethylformamide)-$CH_3CN$ (acetonitrile) organic compound and studied their microstructure [19]. As shown in Figure 6a,b, the surface morphology of the AlCrCuFeMnNi film is homogeneous, with a uniform distribution of spherical particles, but the crack expands significantly with the increase of deposition voltage, indicating that the internal stress of the film increases. The structure of AlCrFeMnNi film is mainly composed of micro-sized flake particles and a few spherical formations, but the film is also peppered with more obvious cracks (Figure 6c,d). Yoosefan et al. studied the effect of deposition parameters on the microstructure of CoCrFeMnNi HEA films [64]. The results show that the surface morphology of CoCrFeMnNi HEA films depends on various pulse parameters, such as duty cycle and frequency, but all of them have different degrees of shrinkage cavity and microcrack. It also indicates that the application of HEA films prepared by electrochemical deposition in the field of tribology has yet to be developed.

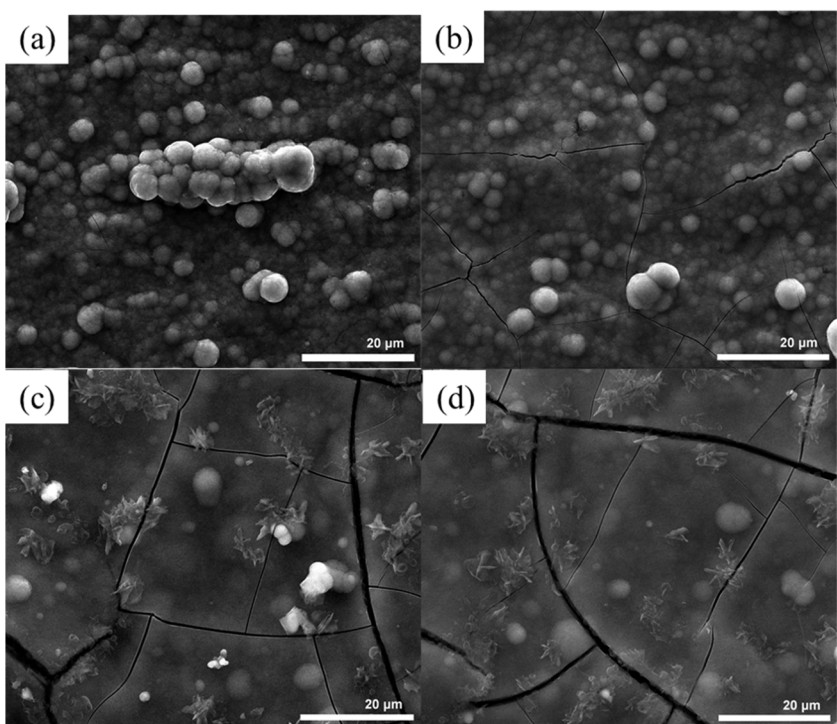

**Figure 6.** SEM images of the HEA films deposited for 90 min: AlCrCuFeMnNi at (**a**) −2.5 V and (**b**) −2.7 V; AlCrFeMnNi at (**c**) −2.5 V and (**d**) −2.7 V [19].

### *3.4. HEA Nitride, Carbide, and Oxide Films*

Recently, HEA nitride, carbide, and oxide films, which have both metal and ceramic advantages, have attracted much more attention due to their higher hardness, strength, and wear resistance. As can be seen from the above preparation method, it is very easy to prepare such cermet films by magnetron sputtering—$N_2$, $O_2$, and $CH_4$ are introduced in the deposition process to achieve reactive deposition. At present, most of the nitride films prepared by magnetron sputtering have FCC or amorphous structure, and a small part have a BCC structure [65]. Garah et al. studied the effect of nitrogen argon flow ratio ($R_{N_2} = N_2 / Ar + N_2$) on the structure and mechanical properties of AlTiZrTaHf(N) HEA films [66]. X-ray diffraction analyses reveal a transition from amorphous to an FCC single phase by increasing the nitrogen content, as shown in Figure 7. The similar phenomenon that low nitrogen flow film presents an amorphous structure, but high nitrogen flow film is transformed into FCC structure, also appears in TiZrHfNiCuCo, AlCrTiZrHf, and AlCrTiZrV films [67–69].

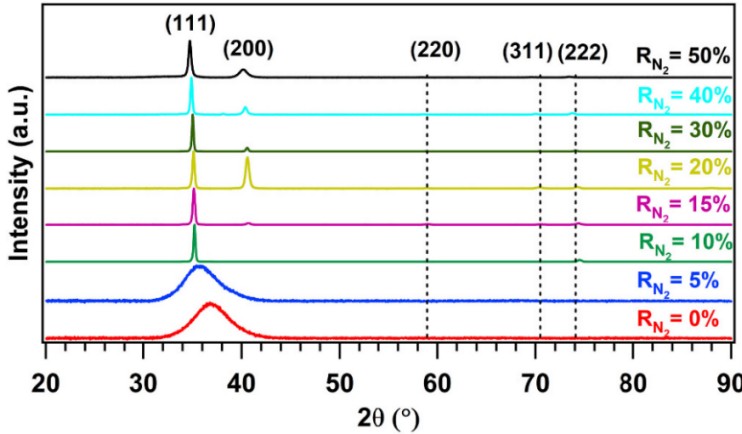

**Figure 7.** XRD patterns of the AlTiTaZrHf(N) HEA films in function of $R_{N2}$ [66].

As can be seen in Figure 8, the film is relatively dense when $R_{N2}$ = 0%, which is related to the higher number of ion channels when the deposition pressure is lower. With the increase of $R_{N2}$, the columnar structure appears in the films corresponding to the FCC structure from the XRD patterns. The density of $R_{N2}$ = 40% is significantly higher than that of the columnar crystal at $R_{N2}$ = 10% because the mobility of the growing plane increases with the increase in nitrogen flow rate [66,70]. Of course, there are other different situations. Sha et al. studied the effect of nitrogen flow rate on the microstructure of FeMnNiCoCr HEA thin film [71]. The thin film always presents a columnar structure, and the higher the nitrogen flow rate, the more the nitrogen content, leading to the transformation of FCC structure to BCC structure. At the same time, other process parameters such as substrate bias and temperature can also affect the microstructure of nitride layer, and higher performance can be achieved by combining with the characteristics of nitride itself. Xu et al. prepared (AlCrTiVZr)N films with different grain sizes and preferred orientations by changing the bias voltage on the substrate [72]. With the increase of bias voltage from 0 to −200 V, the ion bombardment effect increases continuously, the grain size decreases from 23.3 to 11.3 nm, the surface roughness decreases from 1.5 to 0.4 nm, and the residual compressive stress also increases. By observing the cross-section of the thin film, it can be found that the thin film changes from a loose columnar structure to a dense featureless structure. This change, combined with the high hardness of the nitriding layer itself, finally leads to the ultra-high hardness of the material as high as 48.3 GPa. Interestingly, a preference orientation change from (200) to (111) was also observed during the study. Since it has been proved that the (AlCrTiVZr)N films have a NaCl-Type FCC structure, their lowest surface energy plane and the lowest strain energy plane are (200) and (111). The fact that the change of preferred orientation is determined by the strain energy and surface energy, according to the Palleg and others study, suggests that the strain energy and surface energy have a competing relationship, and the minimum total energy of the material orientation party decision, when there is low surface energy, causes the cubic nitride (200) surface texture to grow, and when the strain energy is small, it causes the (111) texture growth [73].

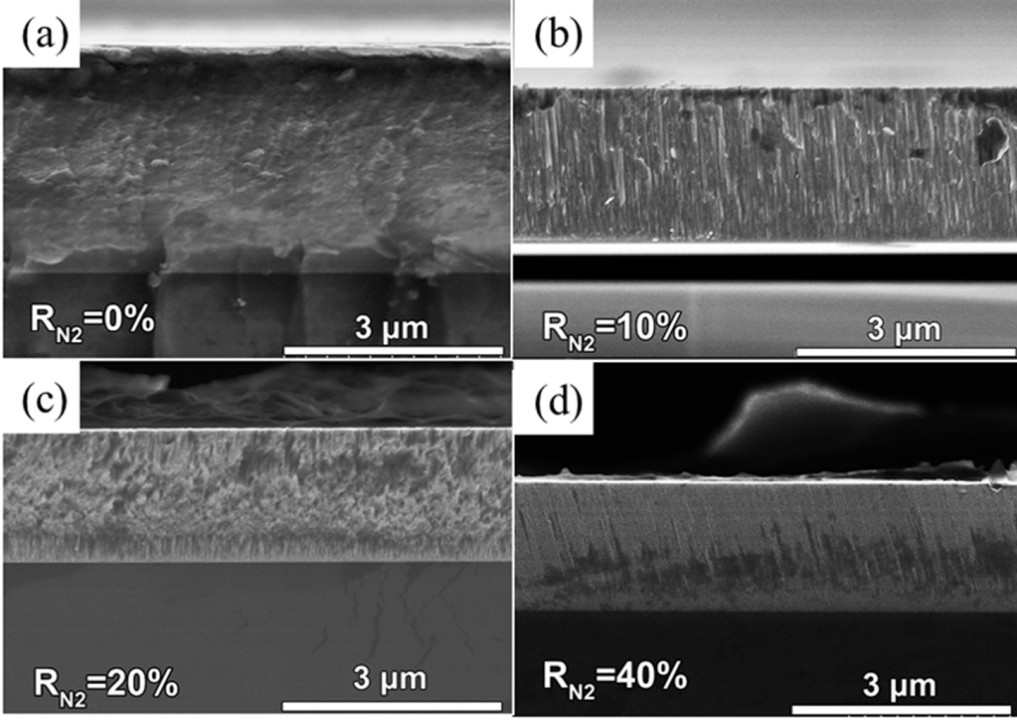

**Figure 8.** Cross-sectional SEM image of AlTiTaZrHf(N) films as a function of $R_{N2}$: (**a**) $R_{N2}$ = 0%; (**b**) $R_{N2}$ = 10%; (**c**) $R_{N2}$ = 20%; (**d**) $R_{N2}$ = 40% [66].

Malinovskis et al. deposited (CrNbTaTiW)C HEA carbide films by non-reactive DC magnetron sputtering and found their structure to be cubic (or distorted cubic) by XRD [74]. Jhong et al. prepared (CrNbSiTiZr)C$_x$ HEA carbide films with different carbon content by reactive radio frequency magnetron sputtering and found that the films had a simple FCC structure [75]. With the increase of carbon content, the XRD peak intensity decreased, resulting in amorphous XRD patterns. Lee et al. prepared CoCrFeNi HEA films by co-deposition and annealed at 1 h for 1273 °C to obtain the film with dispersed oxide particles inside [76]. Analytical STEM imaging found that the particles are Cr$_2$O$_3$, which increased the hardness of the film by 14%. Zhao et al. prepared (FeCrCoNiAl$_{0.1}$)O$_x$ HEA films with different oxygen content by reactive DC magnetron sputtering and found that when the oxygen flow rate was lower than 70%, the film presented a FCC solid solution, and when the oxygen flux reached 80%, the film presented a spinel-type structure [77]. NiCo alloy was precipitated from the HEA films after vacuum annealing at 600 °C.

## 4. Tribological Behavior

Based on the above analysis about HEA coatings and films, it can be seen that laser cladding, thermal spraying, and magnetron sputtering are more conducive to apply in tribology. Therefore, in this section, we introduce the tribological properties of coating/film obtained by the following three preparation methods. Generally, tribology properties of materials are expressed by wear resistance and friction coefficient. According to Archard's theory, the wear rate of materials is inversely proportional to their hardness, while for the coatings/films, the interface bonding force also plays a magnificent role [78]. Besides, the forming degree of the lubricating phase by the chemical reaction of the contact interface can greatly affect friction coefficient. In recent years, various material systems and processes have been tried by researchers to enhance the tribological properties of materials and to adapt to the motion requirements of practical industrial production and application environments. These methods could fall into the following three categories: material design, the addition of a second phase, and process improvement.

### 4.1. Tribological Behavior of Laser Cladding Coatings

One of the ways to improve the tribological properties of HEA coating is to change the structure of solid solution or promote its transformation through alloy composition design to form a solid solution with higher strength and better plastic deformation resistance. Changing the FCC to BCC structure or generating other structures to improve the hardness of the coating and further enhance its wear resistance, Li et al. prepared Al$_x$CrFeCoNiCu coatings with different Al contents by laser cladding, and the structure changed from FCC1 to FCC1 + BCC1 and then BCC1 + BCC2 + FCC2 with Al addition [79]. The hardness of the coating increased from CrFeCoNiCu (215 HV) to Al$_{1.5}$CrFeCoNiCu (625 HV), and the wear rate decreased from $3.50 \times 10^{-4}$ to $6.64 \times 10^{-7}$ mm$^3$ N$^{-1}$ m$^{-1}$. Gu et al. prepared a CoCr$_{2.5}$FeNi$_2$Ti$_x$ coating with different Ti contents by laser cladding and studied its microstructure and wear resistance [80]. The phase structures of the coatings are composed of BCC (Ti = 0) and the solid composition of BCC + FCC phases with the addition of Ti due to the high entropy effect. As shown in Figure 9, the hardness of the coating increases with the increase of Ti content, and the wear weight loss decreases. Shu et al. adjusted the ratio of Fe-Co in the FeCoCrBNiSi laser cladding coating and found that when the ratio was 1:1, the atomic packing density and mixing entropy in the material were large, the glass-forming ability was high, and the hardness and wear resistance of the coating were optimal [81]. Due to the existence of high entropy effect, the change of a certain element type or the increase or decrease of the content in the alloy may lead to the transformation of the structural phase and the emergence of materials with very different properties. This phenomenon is of great significance for obtaining excellent properties of coatings.

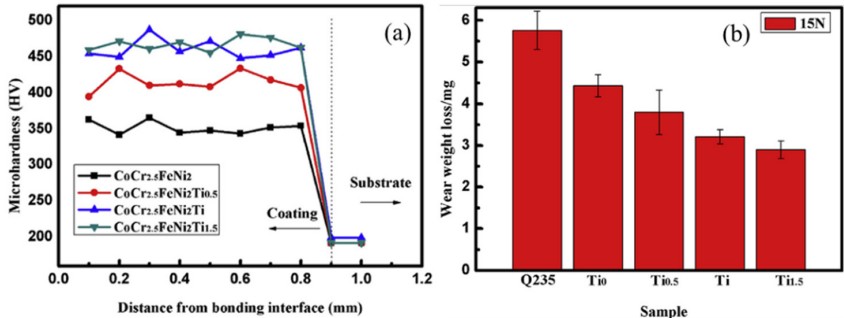

**Figure 9.** (**a**) The micro-hardness of $CoCr_{2.5}FeNi_2Ti_x$ (x = 0, 0.5, 1, 1.5) HEA coatings, and (**b**) the wear loss of the worn samples [80].

Creating ceramic particles by adding non-metallic elements can play a role of dispersive strengthening, which is also a method to improve the coating performance. Liu et al. prepared an AlCoCrFeNiSix laser cladding coating by tuning the content of Si element [82]. With the increase of Si element, more dissolved Si atoms result in severer lattice shrinkage and more precipitating $Cr_{23}C_6$ phase at the grain boundary, acting a stronger pinning effect. When x = 0.5, the highest hardness of 8.19 GPa and the best tribological performance (friction coefficient 0.275; wear rate $1.45 \times 10^{-4}$ mm$^3$ N$^{-1}$ m$^{-1}$) are obtained. Cheng et al. successfully fabricated the $Fe_{25}Co_{25}Ni_{25}(B_xSi_{1-x})_{25}$ laser cladding coating and found that increasing the ratio of B/Si can effectively boost the creation of (Fe, Co, Ni)$_3$B phase and improve the coating hardness [83]. Meanwhile, the wear resistance of coating is also proportional to the x value, as the increasing (Fe, Co, Ni)$_3$B phase with B content improves the surface hardness of coating, enhances the shear resistance of contact surface, and reduces adhesion wear, hence decreasing the wear volume.

As for the adjustment of alloy composition, it can be done by directly adding a lubricating phase, such as graphene and Ag, or producing a certain lubricating phase, such as $Al_2O_3$, $V_2O_5$, $MoO_2$, etc., through wear process to reduce friction coefficient and increase wear resistance. [84,85]. Jin et al. studied the tribological properties of the FeNiCoAlCu HEA coating at different temperatures [86]. The results showed that the coating had no new phase formation and transformation under 400 °C and showed no difference in friction coefficient. However, when the temperature rose to 600 °C, the friction coefficient decreased significantly, and as the temperature continued to rise, the wear stability stage would appear faster, and the stability would be improved obviously, as shown in Figure 10. This phenomenon may be due to the formation of oxide films on the coating surface.

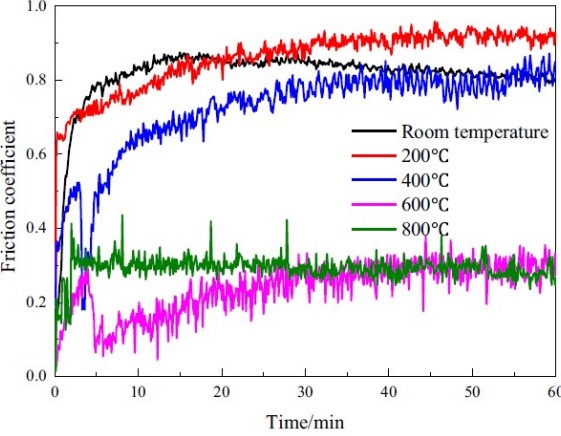

**Figure 10.** Friction coefficient of the FeNiCoAlCu HEA coating at different temperatures [86].

Compared with the adjustment of alloying elements, the direct addition of the second phase is a more controllable way of strengthening because it directly controls the content of the strengthening phase. Zhang et al. mixed FeNiCoCrTi$_{0.5}$ powder with diamond powder

and directly prepared the coating by laser cladding [87]. The results show that the dilution rate of $(FeNiCoCrTi_{0.5})C_x$ coating decreases as the diamond content increases, and the increase of diamond content promotes the generation of carbides ($Cr_{23}C_6$, TiC), which further improves the hardness of the alloy. However, the hardest coating is not the most wear resistant. Due to high hardness and poor plastic shape, the coating with the highest diamond content will have a brittle fracture in the friction process, leading to its wear resistance decline. Similar phenomena have also been observed in the $CoCrCuFeNiSi_{0.2}(TiC)_x$ [88]. Of course, there are also cases that conform to Archard's law. The FeCoCrAlCu(TiC) coating prepared by Jiang et al. increased hardness and wear resistance with increasing TiC content [89].

The selection of coating parameters, improvement of coating microstructure, and surface quality post-treatment are also key to improving the coating performance. The most common method of laser cladding is to change the laser power. Shu et al. prepared CoCrBFeNiSi HEA coatings on H13 steel with different laser powers [90]. The amorphous content in different coatings decreased from 81.15% to 33.79% as the laser power increased from 233 to 700 W. A large amount of the amorphous phase provides a higher coating hardness, and the amorphous phase prevents the further deepening of furrows in the friction process, reduces the adhesion wear, and enhances the wear resistance of the coating. Heat treatment is also a common post-treatment. By reheating to a certain temperature and holding for a period of time, the precipitation in the supersaturated solid solution can be re-precipitated, which can release the residual stress, improve the microstructure, and precipitate the target hardness. Liu et al. annealed $AlCoCrFeNiTi_{0.8}$ prepared by laser cladding at different temperatures and found no significant changes in the microstructure below 700 °C [91]. The coarsening behavior of AlNi precipitates can be observed after heat treatment at 900 °C, and the precipitates present an Ostwald ripening phenomenon after heat treatment at 1200 °C. However, these heat treatments did not improve the wear resistance of high entropy alloy coatings. Cui et al. sulfurized $CoCrFeNiSi_{0.4}$ and CoCrFeMoNi alloys with high entropy prepared by laser cladding and found that the friction coefficients of $CoCrFeNiSi_{0.4}$ and CoCrFeMoNi HEAs were decreased from 0.6 and 0.47 to 0.152 and 0.12 [92]. This is due to the generation of FeS and $MoS_2$ lubrication phase, and the lubrication effect of the two has a synergistic effect in CoCrFeMoNi to further reduce the friction coefficient.

Table 2 shows the main microstructure and tribological properties of HEA coatings prepared by laser cladding in recent years, which can be used as a reference for researchers to develop material systems to a certain extent.

**Table 2.** The microstructure and tribological properties of HEA coatings prepared by laser cladding.

| Coatings | Main Phases | Hardness | Wear Rate/ ($\times 10^{-5}$ mm$^3$ N$^{-1}$ m$^{-1}$) | Friction Coefficients | Design Approach | Ref. |
|---|---|---|---|---|---|---|
| $AlCoCrFeNiSi_x$ | BCC | 8.19 GPa | 14.5 | 0.275 | composition design | [82] |
| $AlCoCrFeNiTi_{0.8}$ | BCC | ~ | 0.164 | 0.52 | Heat treatment | [91] |
| $Al_xCrFeCoNiCu$ | FCC + BCC | 625 HV | 0.0664 | 0.57 | composition design | [79] |
| $Al_xMo_{0.5}NbFeTiMn_2$ | BCC | 1098.5 HV | ~ | 0.40 | composition design | [93] |
| $CoCr_{2.5}FeNi_2Ti_x$ | BCC + FCC | 472 HV | ~ | ~ | composition design | [80] |
| CoCrBFeNiSi | Amorphous | ~1000 HV | ~ | 0.14 | process improvement | [90] |
| $CoCrCuFeNiSi_{0.2}(TiC)_x$ | FCC + TiC | 498.5 HV | ~ | 0.60 | addition of a second phase | [88] |
| $CoCrFeNi_2V_{0.5}Ti_x$ | BCC | 960 HV | 4.43 | ~ | composition design | [94] |
| CoCrFeNiMn | FCC + TiN | ~200 HV | ~ | 0.66 | addition of a second phase | [95] |
| $CoCrFeNiSi_{0.4}$ | FCC | ~640 HV | ~ | 0.60 | process improvement | [92] |
| CoCrFeMoNi | FCC | ~640 HV | ~ | 0.47 | process improvement | [92] |
| $Fe_{25}Co_{25}Ni_{25}(B_{0.7}Si_{0.3})_{25}$ | FCC + BCC + Laves | 11.9 GPa | 0.142 | ~ | composition design | [96] |
| $Fe_{25}Co_{25}Ni_{25}(B_xSi_{1-x})_{25}$ | FCC + Amorphous | 8.39 GPa | ~ | ~ | composition design | [83] |
| (FeCoCrNi-Mo)C | FCC | 600 HV | ~ | 0.19 | addition of a second phase | [97] |
| FeCrCoAlMn0.5Mo0.1 | BCC + FCC | 624.1 HV | ~ | 0.36 | process improvement | [98] |
| $(FeNiCoCrTi0.5)C_x$ | BCC | ~800 HV | ~ | ~ | addition of a second phase | [87] |
| $FeNiCoCrTi0.5Nb_x$ | BCC + FCC + Laves | 852.5 HV | ~ | ~ | composition design | [99] |
| $MgMoNbFeTi_2Y_x$ | FCC + BCC | 1046 HV | ~ | ~ | composition design | [100] |
| $MoFe_{1.5}CrTiWAlNb_x$ | BCC + Laves | 910 HV | ~ | 0.52 | composition design | [101] |
| FeCoCrBNiSi | Amorphous | 850 HV | 60 | 0.17 | composition design | [81] |
| FeCoCrAlCu(TiC) | BCC + TiC | 10.82 GPa | ~ | 0.48 | addition of a second phase | [89] |
| $FeCoCrNiMoSi_x$ | FCC | 826 HV | 1.37 | 0.375 | composition design | [102] |

### 4.2. Tribological Behavior of Thermal Spraying Coatings

HEA coatings prepared by thermal spraying are similar to laser cladding, but it is easy to form holes and oxides during spraying. This is a double-edged sword, the formation of oxide can often improve the hardness of the coating or provide lubrication properties of the coating under high-temperature friction, but the formation of holes is often detrimental to wear resistance. Zhang et al. reported the preparation of nanoscale $Al_2O_3$/13 wt% $TiO_2$ reinforced CoCrFeMnNi HEA coatings via plasma spraying on Q235 [103]. With the increase of $Al_2O_3$/$TiO_2$, the porosity of the coating decreases first and then increases, and the wear resistance of the coating is best when the porosity is small, and the added phase is moderate at 10 wt%. As shown in Figure 11a,b, some spalling pits, grooves, and numerous microcracks were observed in oxidation sheets of the worn surface of the CoCrFeMnNi. It could be inferred that the main wear mechanisms of the CoCrFeMnNi HEA coating were oxidation wear, abrasive wear, severe delamination wear, and contact fatigue. Low incidence of delamination, grooves, and fewer microcracks in the oxidation sheets of the worn surface of the 10 wt% added phase composite coating are shown in Figure 11d,e. The coating was smoother and exhibited much less prominent evidence of fatigue, showing the best friction performance. However, the coating with a 20 wt% additive phase is similar to CoCrFeMnNi, because a large number of pores and oxides cause a large number of spalling pits on the coating surface during friction.

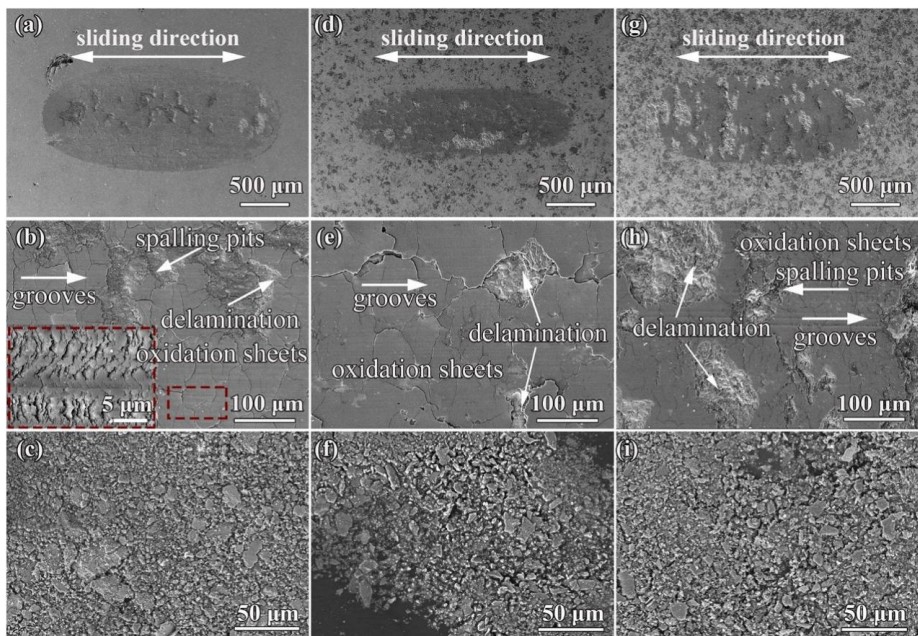

**Figure 11.** SEM images of the worn surface under 50 N: (**a–c**) CoCrFeMnNi HEA, (**d–f**) 10 wt%, (**g–i**) 20 wt% [103].

In addition to the advantages and disadvantages of the coating preparation, the HEA coating prepared by thermal spraying can also be classified from the material design, the addition of a second phase, and process improvement. Zhou et al. prepared $Al_{(1-x)}CoCrFeNiTi_x$ (x = 0, 0.125, 0.25)HEA coatings by HVOF technology and found that with the increase of Ti content, the wear resistance of the coating improved, and the friction coefficient remained stable [104].

Some new phases formed by oxidation of elements during high-temperature friction can significantly reduce friction and wear. Chen et al. studied the tribological properties of $Al_{0.6}TiCrFeCoNi$ HEA coating at different temperatures and found that the friction coefficient decreased by 0.25 when the temperature was 500 °C compared with room temperature, which was due to the oxide film generated by friction reaction, which covered the coating surface at high-temperature friction [29]. Liu et al. studied the tribological

properties of the FeCoCrAlNi HEA coating at room temperature and 800 °C and found that the wear rate of the coating at high temperature was significantly lower than that at room temperature [105]. This is because the friction at room temperature produces a lot of wear debris, forming serious three-body abrasive wear. The coating surface is extremely rough, while the coating at the high-temperature wear track was smooth, with only slight adhesion wear and oxidation wear.

Common additions of the second phase include ceramic particles, such as $Al_2O_3$, which are used to enhance the wear resistance of the coating. Of course, there are also solid lubricants such as Ag to reduce friction. Shi et al. prepared AlCoCrFeNi HEA matrix self-lubricating composite coatings by atmospheric plasma spraying [106]. Studies have shown that the addition of Ag and $BaF_2/CaF_2$ eutectic effectively inhibits abrasion behavior and eminently decreases the wear rate by 10 times. At 800 °C, the addition of Ag and $BaF_2/CaF_2$ eutectic changed the main oxidation products from $Fe_3O_4$, $Cr_2O_3$, and $Al_2O_3$ to $CrO_2$, $Cr_2O_3$, FeO, NiO, and $Al_2O_3$. The appearance of the lubricating phase reduces the friction coefficient of the coating by nearly one time.

The improvement of thermal spraying process and post-treatment can further improve the properties of the coating. The selection of thermal spraying power is the most direct and simple method. Zhang et al. prepared $FeCoCrNiMo_{0.2}$ HEA coating by atmospheric plasma spraying technique and investigated the friction and wear properties of the coating under different spraying powers [107]. The results showed that, with increasing spraying power, the porosity in the coating was reduced due to melting degree increasing, but the content of the oxides in the coating was enhanced. As the spraying power increased from 25.5 to 45 kW, the wear rate decreased, but the friction coefficient had no obvious change. The development of new equipment also provides more options for preparing HEA coatings. Liao et al. prepared a CoCrFeNiMn HEA coating by detonation spraying and found that this method had higher hardness and wear resistance compared with the coating prepared by traditional spraying, and the bonding force between the substrate and the coating was significantly improved [108]. For the coatings prepared by partial spraying technology, which are often carried out in the atmosphere, there will inevitably be residual stress or even a large number of pores or cracks in the preparation process. Laser re-melting technology and subsequent heat treatment are effective measures to solve these problems. Jin et al. prepared $Al_xCoCrFeNiSi$ HEA coatings with different Al contents by plasma spraying and further improved the quality of the coatings by laser re-melting [109]. The results show that the hardness of the coating can reach 1255 HV (x = 2) after laser re-melting. This increase is mainly due to the effects of the precipitation and grain boundary strengthening and the formation of the hard $Cr_3Si$ phase. The coating has a low friction coefficient of 0.25 and an excellent wear rate of $3.2 \times 10^{-7}$ mm$^3$ N$^{-1}$ m$^{-1}$ in friction experiments. Wu et al. annealed the CoCrFeNiMn HEA coating at 600 and 900 °C and found that the hardness of the coating increased after annealing at 600 °C due to the precipitation of oxide, while the hardness of the coating decreased sharply after annealing at 900 °C, which was the result of grain growth [110]. Although the hardness of the coating decreases after annealing at 900 °C, the wear resistance of the coating increases due to the improvement of the bonding force between the coating and the substrate.

Table 3 shows the main microstructure and tribological properties of HEA coatings prepared by thermal spraying in recent years, which can be used for reference.

**Table 3.** The microstructure and tribological properties of HEA coatings prepared by thermal spraying.

| Coatings | Main Phases | Hardness | Wear Rate/ ($\times 10^{-5}$ mm$^3$ N$^{-1}$ m$^{-1}$) | Friction Coefficients | Design Approach | Ref. |
|---|---|---|---|---|---|---|
| $Al_{0.75}CoCrFeNiTi_{0.25}$ | BCC | 8.869 GPa | 7.5 | ~0.7 | composition design | [104] |
| Al0.2CrFeNiCu | FCC | 591 HV | ~ | ~0.2 | process improvement | [111] |
| Al0.2CrFeNiCo | FCC + BCC | 361 HV | ~ | ~0.13 | process improvement | [111] |
| Al0.6TiCrFeCoNi | BCC | 789 HV | 10.44 | ~0.75 | ~ | [29] |
| AlCoCrFeMo | BCC | 5.78 GPa | 54.9 | ~ | process improvement | [112] |

**Table 3.** *Cont.*

| Coatings | Main Phases | Hardness | Wear Rate/ ($\times 10^{-5}$ mm$^3$ N$^{-1}$ m$^{-1}$) | Friction Coefficients | Design Approach | Ref. |
|---|---|---|---|---|---|---|
| AlCoCrFeNi | BCC + Ag + BaF$_2$ + CaF$_2$ | ~500 HV | 2.5 | 0.54 | addition of a second phase | [106] |
| AlCoCrFeNiTi$_{0.5}$ | BCC | 610 HV | ~ | 1.3 | ~ | [113] |
| Al$_x$CoCrFeNiSi | BCC + FCC | 1255 HV | 0.032 | 0.24 | composition design | [109] |
| CoCrFeMnNi | FCC + Al$_2$O$_3$/TiO$_2$ | 7.5 GPa | 0.5 | ~0.4 | addition of a second phase | [103] |
| CoCrFeNiAl | BCC + FCC | 383 HV | 26 | 0.76 | composition design | [114] |
| CoCrFeNiMn | FCC + MnCr$_2$O$_4$ | 359 HV | 3.18 | 0.51 | composition design | [114] |
| CoCrFeNiMo$_{0.5}$ | FCC + Cr$_2$FeO$_4$ | 312 HV | 12.7 | 0.56 | composition design | [114] |
| CoCrFeNiMn | FCC | 471 HV | ~ | ~ | process improvement | [108] |
| CoCrFeNiMn | FCC | 551 HV | 10.2 | ~ | process improvement | [110] |
| CoFeNiCrTaAl | FCC + Laves | 546.6 HV | 4.5 | 0.616 | ~ | [115] |
| FeCoCrAlNi | BCC | 542 HV | 10.2 | 0.27 | ~ | [105] |
| FeCoCrNiMo$_{0.2}$ | FCC + Oxide | 558 HV | 11.8 | 0.78 | process improvement | [107] |
| FeCoNiCrMn(Al$_2$O$_3$) | FCC + Al$_2$O$_3$ | 302 HV | 24 | 0.96 | addition of a second phase | [116] |
| FeCoNiCrSiAl$_x$ | FCC + BCC | 439 HV | 0.67 | 0.6 | composition design | [56] |
| HfNbTaZr | BCC | 9.51 GPa | 59 | 0.33 | process improvement | [117] |
| MoNbTaVW | BCC | 7.69 GPa | 37 | 0.26 | process improvement | [117] |

*4.3. Tribological Behavior of Magnetron Sputtering Films*

Compared with coating, films prepared by magnetron sputtering are more precise and generally have higher microhardness. Moreover, due to its extremely low surface roughness, the initial stage of friction is more stable, and then it is easy to obtain films with good friction properties. Alvi et al. prepared an HEA thin film CuMoTaWV with a thickness of ~900 nm by magnetron sputtering and annealed it to study the tribological properties before and after annealing [118]. As shown in Figure 12a, the as-deposited refractory HEA film showed strong plastic deformation after the room temperature friction test, and the friction coefficient increased from 0.2 to 0.7, suggesting the removal of the film. Although the roughness of the films annealed at 300 °C increased, and the friction coefficient was higher than that of the as-deposited films at the initial stage of friction, the enhancement of the film base bonding force made it difficult to remove off, and it maintained a friction coefficient of 0.25. As shown in Figure 12c, the wear track from the tribological test at 300 °C showed low abrasive wear, showing the stability of the film.

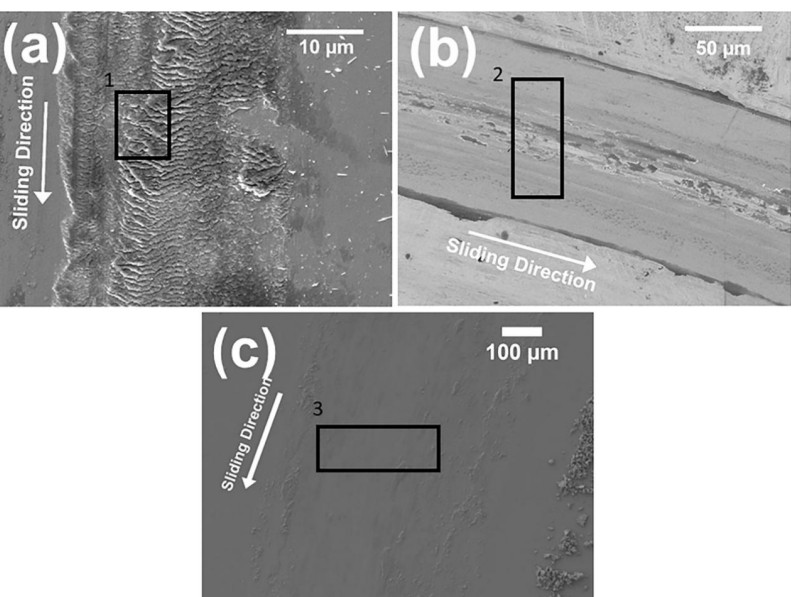

**Figure 12.** SEM images of the worn surface (**a**) as-deposited film, (**b**) 300 °C annealed film, and (**c**) tribotest at 300 °C [118].

Composition control is also applicable to magnetron sputtering deposited films. Zhao et al. regulated the tribological properties of thin films by introducing Mo element into FeCoNiCrMn and found that the friction coefficient decreased gradually with the

addition of the Mo element [119]. With the addition of the Mo element, the film changes from FCC to BCC to improve the hardness of the film, and the film becomes denser and the grain refined, which promotes the improvement of its friction performance. Similar to the previous literature, Fan et al. prepared an HEA film VAlTiCrMo by controlling the content of the Mo element and studied its tribological properties at high temperature [120]. VAlTiCr coating is relatively stable in the process of friction at 700 °C and is typical of abrasive wear and oxidation wear. However, with the addition of Mo, the upward diffusion and segregation of solid solution metals in the HEA films are promoted, and a special layered oxide layer is formed. It is the special oxides like $V_2O_5$, $Al_2(MoO_4)_3$, $AlVO_4$, that dramatically reduce the friction coefficient.

Compared with the HEA coatings, the addition of the second phase is generally ceramic particles, while the addition of the second phase of the film prepared by magnetron sputtering is generally through the adjustment of the layered structure. Luo et al. prepared NbMoWTa/Ag Self-lubricating multilayers with different sublayer thicknesses by magnetron sputtering were studied, and it was found that the film could maintain a high hardness under the condition of adding 50% Ag, and it achieved a good wear-resistant self-lubricating effect [28]. This is because as the sublayer thickness decreases, the transition of the deformation mechanism from classic Hall−Petch strengthening to coherent strengthening, which makes the film have a high load-bearing capacity. As shown in Figure 13, the durable lubrication film formed by the addition of Ag reduces the shear force between the film and the counter body and overcomes the brittleness of NbMoWTa film, which greatly improves its friction performance.

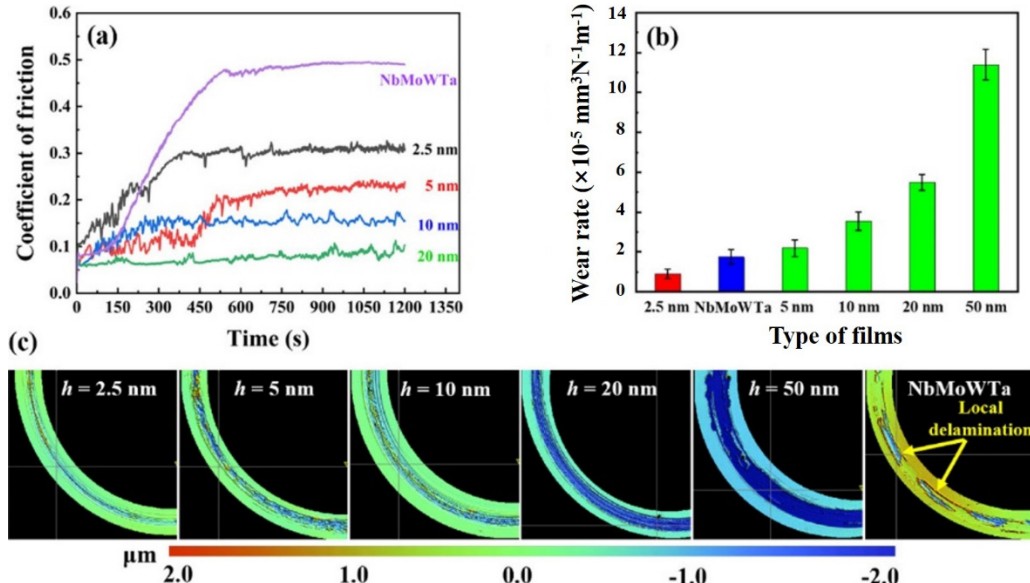

**Figure 13.** The tribological properties and surface profile of all the NbMoWTa/Ag multilayers and monolithic NbMoWTa film. (**a**) COF and (**b**) wear rate of different films. (**c**) White light interferometry images of the worn surfaces [28].

In the process of preparing thin films by magnetron sputtering, the substrate bias voltage is a very important parameter, which can increase the bonding strength of substrate and film and plays a key role in the friction properties of thin films. Wang et al. prepared a series of HEA films CrNbTiMoZr by changing the substrate bias voltage and found that the columnar structure of the films gradually disappeared with the increase of the bias voltage and the wear resistance gradually deteriorated [121]. The authors consider that the deformation mechanism of the thin film is similar to grain boundary sliding, which divides the applied load into the central sliding zone supporting the matrix load and the outer zone supporting the balanced applied load, thus improving the wear resistance of the thin film. Yu et al. also prepared CrNbSiTiZr HEA films by changing the substrate bias

voltage and found that the film was amorphous no matter what the bias voltage was [122]. The series films exhibit adhesive wear and abrasive wear at low bias voltage and abrasive wear at high bias voltage, which is caused by the decrease of hardness.

Due to the existence of Archard's theory, researchers want to improve hardness to meet the requirements of wear resistance, so HEA nitride and carbide films have entered the attention of scholars in the field of friction. In recent years, in order to improve the wear resistance and friction reduction performance of HEA nitride films, researchers have started from the direction of nitrogen flux and invested a lot of energy. Feng et al. prepared a series of $(CrTaNbMoV)N_x$ HEA nitride films by regulating the $N_2/(Ar + N_2)$ flow ratios $(R_N)$ and found that the friction coefficient of nitride films was decreased in different levels compared with alloy film [123]. The nitride films have a lower friction coefficient, which may due to the lower surface roughness compared with the alloy film. Contrary to the previous results, Bachani et al. studied $(TiZrNbTaFe)N_x$ HEA nitride film and found that the friction coefficient of nitride film was all higher than that of alloy film [124]. Cui et al. studied AlCrTiZrHf HEA nitride films and found that, when the $N_2$: Ar ratio was 5:4, the hardness and elastic modulus of the (AlCrTiZrHf)N film reached the maximum, and it also had the lowest friction coefficient [68].

The improvement of preparation technology is also one of the main research directions of HEA nitride thin films. Lo, et al. prepared (AlCrNbSiTiMo)N HEA nitride films by changing the substrate bias and found that the ratio of $H^3/E^2$ gradually increased with the increase of substrate bias [25]. However, due to the influence of residual stress, the wear rate does not decrease with the increase of $H^3/E^2$ ratio, the wear rate reaches the lowest when the substrate bias is $-100$ V. Wang et al. prepared (AlCrWTiMo)N HEA nitride thin films by changing the temperature of the substrate and found that, from the interlayer surfaces to the film surfaces, the films showed gradient structures [125]. With the increase of the substrate temperature, the grain size, surface roughness, and the width of the amorphous region decreased, and the width of the nanocrystalline region increased. When the substrate temperature is 200 °C, the mechanical properties and friction properties of the film reach the best level. Of course, there is the combination of both research studies. Lin et al. prepared $(Cr_{0.35}Al_{0.25}Nb_{0.12}Si_{0.08}V_{0.20})N$ HEA nitride thin films by adjusting substrate temperature and bias and found that the temperature control is beneficial to density and residual stress of the film, and the increase of bias voltage increases point defects, which is beneficial to improve the hardness of the film [126].

A few researchers have also improved the tribological properties of HEA nitride films by other methods. Huang et al. believed that the bonding force of a single-layer HEA nitride film was not strong, and the residual stress was high, so they tried to add a layer of elastic NiTi film between the substrate and nitride film to improve its friction performance [127]. By changing the thickness of NiTi layer, the bonding force of the film is improved, and the residual stress is reduced. When the thickness of NiTi layer is 600 nm, the best friction performance is obtained.

As for the carbide layer of high entropy alloy, Kao et al. [128] prepared a TaNbSiZrCr carbide layer on WC substrates by radio-frequency unbalanced magnetron sputtering and studied the effect of $C_2H_2$ gas flow rate on its microstructure tribological properties. It is found that the increase of carbon content changes the layer into DLC structure, which has a higher H/E ratio, higher $H^3/E^2$ ratio, and lower friction coefficient. When the flow rate of $C_2H_2$ is 23 sccm, the coating has the best tribological performance and wear resistance $(0.33 \times 10^{-6} \text{ mm}^3 \text{ N}^{-1} \text{ m}^{-1})$. The main microstructure and tribological properties of HEA films and HEA nitride and carbide films prepared by magnetron sputtering in recent years are given in Table 4 for reference.

By comparing the above three kinds of HEA wear-resistant coatings, it can be found that each of them has advantages to meet different industrial needs. The wear-resistant coating prepared by laser cladding has a metallurgical combination with the matrix, strong bonding force, controllable melting temperature, and relatively compact coating, which is suitable for harsh environment such as the petrochemical industry [12,13]. The film

prepared by magnetron sputtering has very low roughness and high hardness, which is more suitable for surface modification of precision parts, such as sensors and microelectronics [27,118]. The HEA coating prepared by thermal spraying has high deposition efficiency, low coating dilution, and low cost [16,108]. It is suitable for traditional industrial fields such as all kinds of motor bearings and all kinds of rolls.

**Table 4.** The microstructure and tribological properties of HEA films prepared by magnetron sputtering.

| Films | Main Phases | Hardness | Wear Rate/ ($\times 10^{-5}$ mm$^3$ N$^{-1}$ m$^{-1}$) | Friction Coefficients | Design Approach | Ref. |
|---|---|---|---|---|---|---|
| CuMoTaWV | BCC | 19 GPa | 0.64 | 0.25 | process improvement | [118] |
| NbMoWTa/Ag | BCC | ~9.4 GPa | 0.9 | 0.3 | addition of a second phase | [28] |
| CrNbSiTiZr | Amorphous | 12.4 GPa | ~ | 0.53 | process improvement | [122] |
| CrNbTiMoZr | Amorphous + BCC | 9.7 GPa | ~ | 0.5 | process improvement | [121] |
| VAlTiCrMo$_{0.6}$ | BCC | 10.35 GPa | 11.6 | 0.53 | composition design | [120] |
| VAlTiCr | BCC | 7.3 GPa | 21.5 | 0.55 | composition design | [120] |
| FeCoNiCrMnMo | FCC + BCC | 12 GPa | ~ | 0.3 | composition design | [119] |
| (AlCrNbSiTiMo)N | FCC | 34.5 GPa | 0.12 | 0.68 | process improvement | [25] |
| (Cr$_{0.35}$Al$_{0.25}$Nb$_{0.12}$Si$_{0.08}$V$_{0.20}$)N | FCC | 35 GPa | 0.2 | 0.88 | process improvement | [126] |
| (CrAlTiNbV)N | FCC | 35.5 GPa | ~ | ~ | process improvement | [129] |
| (CrNbTiAlV)N$_x$ | FCC | 49.95 GPa | ~ | ~ | composition design | [130] |
| (CrNbTiAlV)N | FCC | 35.3 GPa | ~ | ~ | process improvement | [131] |
| (CrTaNbMoV)N$_x$ | FCC | 21.6 GPa | 0.84 | ~0.65 | composition design | [123] |
| (AlCrTiZrHf)N | Amorphous + FCC | 33.1 GPa | ~ | 0.5 | composition design | [68] |
| (AlCrWTiMo)N | FCC | 24.5 GPa | 1.1 | 0.67 | process improvement | [125] |
| NiTi /(CrAlNbSiV)N | FCC | 31.1 GPa | ~ | ~0.8 | composition design | [127] |
| (FeMnNiCoCr)N$_x$ | FCC | 17 GPa | 0.018 | ~ | composition design | [71] |
| (TiZrNbTaFe)N$_x$ | Amorphous | 36.2 GPa | 0.075 | 0.69 | composition design | [124] |
| (TaNbSiZrCr)C$_x$ | Amorphous + FCC | 14.4 GPa | 0.033 | 0.09 | composition design | [128] |
| (CrNbSiTaZr)C$_x$ | Amorphous | 14 GPa | 0.016 | 0.05 | composition design | [132] |
| (CrNbSiTiZr)C$_x$ | Amorphous | 32 GPa | 0.02 | 0.07 | composition design | [75] |

Based on the above review of HEA coating/film, it can be seen that the HEA has better wear-resistance and friction-reduction performance, which can be attributed to the following four points: First, HEA coatings/films generally have high hardness and strength, which is exactly what wear-resistant materials need. Second, the strong strengthening effect of solid solution and the second-phase strengthening caused by hard compounds in high entropy alloy effectively enhance the ability of the coating to resist plastic deformation. Thirdly, the presence of a large number of nanocrystals or amorphous in the alloy optimizes the microstructure and properties of the material. Fourth, the residual compressive stress generated during the preparation of coating and film contributes to its tribological properties. Fifth, HEA has a wide variety of elements, such as Mo, V, W, and so on, which may generate lubricating film in the process of friction, helping to reduce the friction coefficient.

## 5. Conclusions and Suggestion

HEA coatings/films have been explored and studied for almost 10 years. In this paper, laser cladding, thermal spraying, and magnetron sputtering, which display advantages for film deposition in the field of tribology, are discussed, and some research achievements are listed. It can be seen that with the guidance of strengthening criterion in HEA, many advanced film systems with excellent tribological properties have been discovered. However, the theoretical understanding and computer simulation for the guidance of HEA film research is not enough. On the other hand, standardized production, large-scale industrial application, and cost consumption for HEA film are still the urgent problems to be solved. Therefore, future research directions can focus on the following aspects:

(1) The literature on modeling and simulation of the tribological properties of HEA coatings and films are very limited. A more atomistic understanding is helpful to elucidate the tribological mechanism of HEA and guide the development of new HEA structures with superior wear resistance and friction reduction.

(2) There are few studies on the tribological properties of HEA composite coatings/films. Through the interface structure design, materials with different performance characteristics can be combined with each other, which can further improve the strength

and toughness of materials and obtain more fatigue-resistant and stable HEA coatings/films.

(3) The research on self-lubricating system of HEA is far less than that on enhancing its wear resistance. Therefore, the metal self-lubricating material system can be further expanded by referring to the experience of traditional self-lubricating composite materials, e.g., incorporating Ag, two-dimensional material graphene, and metal compound $MoS_2$, etc.

(4) The tribological performance of HEA films in severe environments needs clarification, such as high/low-temperature, corrosion, vacuum, and other conditions, the data accumulation of which can promote the practical application of HEA coating/films in various environments.

(5) It is necessary to develop new preparation technology that is more suitable for large-scale industrial production and to promote the development of customized performance and production automation of HEA coatings and films in practical applications.

**Author Contributions:** Conceptualization, D.L. and Q.Z.; methodology, Z.H.; validation, Y.L. (Yulong Li) and Y.L. (Yulin Liu); formal analysis, Q.L.; investigation, Y.H.; resources, H.W.; data curation, D.L.; writing—original draft preparation, D.L.; writing—review and editing, D.L.; supervision, Q.Z. All authors have read and agreed to the published version of the manuscript.

**Funding:** This research was funded by National Natural Science Foundation of China (No. 52175188, 51975474), State Key Laboratory for Mechanical Behavior of Materials (20222412) and the Fundamental Research Funds for the Central Universities (3102019JC001).

**Institutional Review Board Statement:** Not applicable.

**Informed Consent Statement:** Not applicable.

**Data Availability Statement:** All data analyzed in this study are included in this published article.

**Conflicts of Interest:** The authors declare no conflict of interest.

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
