# Peer review of "Tribological Behavior of High Entropy Alloy Coatings: A Review"

_coatings, doi:10.3390/coatings12101428_

Round 1

Reviewer 1 Report

1.      The article has a interesting work for the journal reader and for scientific community. It is advised that the manuscript must be read carefully and remove all typographical mistakes. The grammatical mistakes and sentence sequences. There are some mistakes in the graphs captions and also in the graphs text. Remove all these mistakes.

2.      The Discussion section must be rewritten to be better understood by a broad audience and the conclusions in the manuscript should be more organized. Please provide the implications of your results. A depth discussion on the tribological behavior of high entropy alloy coatings would bring more value to your paper.

3.      Please provide a brief introduction and motivation for the conducted study.

4.      There aren’t appropriate and adequate references to related and previous work in this paper.

5.      Authors are advised to extend a little bit the presented research literature. There are some papers which falls into the scope of the article, which may be checked (10.1016/j.surfcoat.2021.128009,10.1007/s11666-022-01350-y, 10.3390/ma14195814).

6.      Some parts of paper should be presented in a briefer and more concise way. It reads a little like a thesis and not like a research article.

Reviewer 2 Report

The submitted manuscript “Tribological behavior of high entropy alloy coatings: A Review” definitely provides valuable information. The author reported here how the evolution of the HEA coatings took place over decades considering tribological applications.

The article is focused to the topic. There are few queries for authors to address. Having done those, the manuscript can be accepted for publication in the journal. Authors should include all the clarifications and responses to the queries in the revised manuscript.

Comments:

1.     Author needs to rectify the direction of the arrows in Figure 1 to properly indicate the equiaxed and columnar grains.

2.     It will be quite interesting for the readers if the author can illustrate through a graphical representation that how HEA coatings got evolved over the years in the Introduction section.

3.     “…… the parameters of mixing entropy and mixing enthalpy, ?, …”. Please define the term ‘?’ before mentioning it. Page 5, Line 197.

4.     “With the decrease of VEC, the crystal structure of the alloy changes from FCC to BCC. When VEC<6.8, the solid solution is FCC structure; otherwise, it is BCC structure.” Please clarify and rewrite the statements. Page 6, Line 225-227.

5.     Why the XRD peak at (200) is so strong compared to others at RN2= 20% in Figure 7?

Reviewer 3 Report

No Comments.

Author Response

Thanks the reviewer for his/her recognition of our effort.

Reviewer 4 Report

The manuscript presents a review of the tribological properties and behavior of high-entropy alloys in the form of films and coatings deposited by different techniques. Overall, the paper is interesting and can be useful for the scientific community. The authors should pay attention to the following aspects:

- Although the advantages of the HEAs are discussed in the introduction, the applications of these materials in the industry should be mentioned.

- The authors discussed the techniques for the preparation of HEAs coatings in paragraph 3. However, the advantages and disadvantages of these methods are discussed very poorly. This information will lead to a better understanding of the basics of the deposition of the discussed coatings and films. 

- The authors presented a specific approach for the prediction of the phase structure. However, the information related to the correctness, advantages, and disadvantages of the proposed method is missing. The authors should add this information. 

- In my opinion, paragraphs 4.2 and 4.3 should be renamed - "Friction and wear" should be replaced by "Tribological behavior" due to the title of paragraph 4.  Also, the title of subparagraph 4.1. needs correction by adding the word "behavior".

- The authors should make a comparison between the functional properties of the considered coatings and films formed by the different approaches, i.e. laser cladding, thermal spraying, etc., and discuss which technology is more appropriate for a certain practical application.

Reviewer 5 Report

Dear Authors,

In my opinion, this is a good paper dealing with a very topical subject. The paper is interesting and well written; the text is clear and easy to read.

I liked the article; for me and future readers it will be a good compendium of the subject matter presented. My comments only concern an editorial issue, please pay attention to the subscripts, in many places they are not included (e.g. page 13 last paragraph, page 17 paragraph under the table, etc.).

Furthermore, the title of Chapter 4.1 is incomplete, it should be: Tribological Properties of laser cladding coatings; the title of Chapter 4.3 should start with a capital letter

Round 2

Reviewer 1 Report

The authors have revised the paper very well.